# Agnostic $Q$-learning with Function Approximation in Deterministic Systems: Near-Optimal Bounds on Approximation Error and Sample Complexity

**Simon S. Du**[*]      Jason D. Lee[†]      Gaurav Mahajan[‡]      Ruosong Wang[§]

## Abstract

The current paper studies the problem of agnostic $Q$-learning with function approximation in deterministic systems where the optimal $Q$-function is approximable by a function in the class $\mathcal{F}$ with approximation error $\delta \geq 0$. We propose a novel recursion-based algorithm and show that if $\delta = O\left(\rho/\sqrt{\dim_E}\right)$, then one can find the optimal policy using $O(\dim_E)$ trajectories, where $\rho$ is the gap between the optimal $Q$-value of the best actions and that of the second-best actions and $\dim_E$ is the Eluder dimension of $\mathcal{F}$. Our result has two implications:

1.  In conjunction with the lower bound in [Du et al., 2020], our upper bound suggests that the condition $\delta = \widetilde{\Theta}\left(\rho/\sqrt{\dim_E}\right)$ is necessary and sufficient for algorithms with polynomial sample complexity.
2.  In conjunction with the obvious lower bound in the tabular case, our upper bound suggests that the sample complexity $\widetilde{\Theta}\left(\dim_E\right)$ is tight in the agnostic setting.

Therefore, we help address the open problem on agnostic $Q$-learning proposed in [Wen and Van Roy, 2013]. We further extend our algorithm to the stochastic reward setting and obtain similar results.

## 1 Introduction

$Q$-learning is a fundamental approach in reinforcement learning [Watkins and Dayan, 1992]. Empirically, combining $Q$-learning with function approximation schemes has lead to tremendous success on various sequential decision-making problems. However, theoretically, we only have a good understanding of $Q$-learning in the tabular setting. Strehl et al. [2006], Jin et al. [2018] show that with certain exploration techniques, $Q$-learning provably finds a near-optimal policy with sample complexity polynomial in the number of states, number of actions and the planning horizon. However, modern reinforcement learning applications often require dealing with large state space where the polynomial dependency on the number of states is not acceptable.

Recently, there has been great interest in designing and analyzing $Q$-learning algorithms with linear function approximation [Wen and Van Roy, 2013, Du et al., 2019]. Under various additional assumptions, these works show that one can obtain a near-optimal policy using $Q$-learning with sample complexity polynomial in the feature dimension $d$ and the planning horizon, if the optimal $Q$-function is an exact linear function of the $d$-dimensional features of the state-action pairs.

---

[*]University of Washington. Email: `ssdu@cs.washington.edu`

[†]Princeton University. Email: `jasonlee@princeton.edu`

[‡]University of California, San Diego. Email: `gmahajan@eng.ucsd.edu`

[§]Carnegie Mellon University. Email:`ruosongw@andrew.cmu.edu`

A major drawback of these works is that the algorithms can only be applied in the well-specified case, i.e., the optimal $Q$-function is an exact linear function. In practice, the optimal $Q$-function is usually linear up to small approximation errors instead of being exactly linear. In this paper, we focus on the agnostic setting, i.e., the optimal $Q$-function can only be approximated by a function class with approximation error $\delta$, which is closer to practical scenarios. Indeed, designing a provably efficient $Q$-learning algorithm in the agnostic setting is an open problem posed by Wen and Van Roy [2013].

Technically, the agnostic setting is arguably more challenging than the exact setting. As recently shown by Du et al. [2020], for the class of linear functions, when the approximation error $\delta = \Omega(\sqrt{\mathrm{poly}(H)/d})$ where $H$ is the planning horizon, any algorithm needs to sample exponential number of trajectories to find a near-optimal policy even in deterministic systems. Therefore, for algorithms with polynomial sample complexity, additional assumptions are needed to bypass the hardness result. For the exact setting $\delta = 0$, Wen and Van Roy [2013] show that one can find an optimal policy using polynomial number of trajectories for linear functions in deterministic systems, which implies that the agnostic setting could be exponentially harder than the exact setting.

Due to the technical challenges, for the agnostic setting, previous papers mostly focus on the bandit setting or reinforcement learning with a generative model [Lattimore and Szepesvari, 2019, Van Roy and Dong, 2019, Neu and Olkhovskaya, 2020], and much less is known for the standard reinforcement learning setting. In this paper, we design $Q$-learning algorithms with provable guarantees in the agnostic case for the standard reinforcement learning setting.

## 1.1 Our Contributions

Our main contribution is a provably efficient $Q$-learning algorithm for the agnostic setting with general function approximation in deterministic systems. Our results help address the open problem posed by Wen and Van Roy [2013].

**Theorem 1.1** (Informal). *For a given episodic deterministic system and a function class $\mathcal{F}$, suppose there exists $f \in \mathcal{F}$ such that the optimal $Q$-function $Q^*$ satisfies $|f(s,a) - Q^*(s,a)| \leq \delta$ for any state-action pair $(s,a)$. Suppose $\rho = \Omega(\sqrt{\dim_E}\delta)$, where the optimality gap $\rho$ is the gap between the optimal $Q$-value of the best action and that of the second-best action (formally defined in Definition 3.1) and $\dim_E$ is the Eluder dimension of $\mathcal{F}$ (see Definition 3.5), our algorithm finds the optimal policy using $O(\dim_E)$ trajectories.*

Our main assumption is that the optimality gap $\rho$ satisfies $\rho = \Omega(\sqrt{\dim_E}\delta)$. Below we discuss the necessity of this assumption and its connection with the recent hardness result in Du et al. [2020].

In Du et al. [2020], it has been proved that in deterministic systems, if the optimality gap $\rho = 1$ and the optimal $Q$-function can be approximated by linear functions with approximation error $\delta = \Omega(\sqrt{\mathrm{poly}(H)/d})$, any algorithm needs to sample exponential number of trajectories to find a near-optimal policy even in deterministic systems. Here $d$ is the input dimension of the linear functions. Using the same technique as in [Du et al., 2020], we show the following hardness result for $Q$-learning with linear function approximation in the agnostic setting.

**Proposition 1.2** (Generalization of Theorem 4.1 in [Du et al., 2020]). *For any $\rho \leq 1$, there exists a family of deterministic systems where the optimality gap is $\rho$ and the optimal $Q$-function can be approximated by linear functions with approximation error $\delta = O(C/\sqrt{d} \cdot \rho)$, such that any algorithm that returns a $\rho/2$-optimal policy needs to sample $\Omega(2^C)$ trajectories.*

By setting $C = O(\log(Hd))$ such that $2^C = \mathrm{poly}(Hd)$, Proposition 1.2 implies that for *any* algorithm with polynomial sample complexity, the approximation error $\delta$ that can be handled is at most $\widetilde{O}(\rho/\sqrt{d})$. Recall that the Eluder dimension of linear functions is $\widetilde{O}(d)$. Theorem 1.1 suggests that as long as $\rho = \tilde{\Omega}(\sqrt{d}\delta)$, our algorithm finds the optimal policy using polynomial number of samples. Note that this applies to every pair of $(\rho, \delta)$ that satisfies the condition. Proposition 1.2 suggests that there exist environments with $\rho = \tilde{\Omega}(\sqrt{d}\delta)$ which require exponential number of samples to find a near-optimal policy. Therefore, combining Theorem 1.1 and Proposition 1.2, we give a tight characterization (up to logarithmic factors) on the quantitative relation between $\rho$ and $\delta$ under which one can use polynomial number of samples to find the optimal policy.

Our result is in the same spirit as the results in [Lattimore and Szepesvari, 2019, Van Roy and Dong, 2019], which also demonstrate the tightness of the hardness result in [Du et al., 2020].

However, as will be made clear, technically our result significantly deviates from those in [Lattimore and Szepesvari, 2019, Van Roy and Dong, 2019]. See Section 2 for more detailed comparison with [Lattimore and Szepesvari, 2019, Van Roy and Dong, 2019].

Note that the sample complexity of our algorithm is linear in the Eluder dimension of the function class. For the tabular setting, the Eluder dimension is as large as the cardinality of the state-action space [Russo and Van Roy, 2013]. This cardinality is also a sample complexity lower bound, i.e., for the tabular setting, the sample complexity lower bound is $\Omega\left(\dim_E\right)$. Therefore, our sample complexity is also tight.

Finally, we show how to generalize our results to handle stochastic rewards. Under the same assumption that $\rho = \Omega(\sqrt{\dim_E}\delta)$, our algorithm finds an optimal policy using $\frac{\text{poly}(\dim_E, H)}{\rho^2}\log(1/p)$ trajectories with failure probability $p$. We would like to remark that the $\log(1/p)/\rho^2$ dependency is necessary for finding optimal policies even in the bandit setting [Mannor and Tsitsiklis, 2004].

**Organization**   In Section 2, we review related work. In Section 3, we introduce necessary notations, definitions and assumptions. In Section 4, we discuss the special case where $\mathcal{F}$ is the class of linear functions to demonstrate the high-level approach of our algorithm and the intuition behind the analysis. We then present the result for general function classes in Section 5. We conclude in Section 6 and defer proofs to the supplementary material.

## 2   Related Work

Classical theoretical reinforcement learning literature studies asymptotic behavior of concrete algorithms or finite sample complexity bounds for $Q$-learning algorithms under various assumptions [Melo and Ribeiro, 2007, Zou et al., 2019]. These works usually assume the initial policy has certain benign properties, which may not hold in practical applications. Another line of work focuses on sample complexity and regret bound in the tabular setting [Lattimore and Hutter, 2012, Azar et al., 2013, Sidford et al., 2018a,b, Agarwal et al., 2019, Jaksch et al., 2010, Agrawal and Jia, 2017, Azar et al., 2017, Kakade et al., 2018]. Strehl et al. [2006], Jin et al. [2018] show that with certain exploration techniques, $Q$-learning provably finds a near-optimal with polynomial sample complexity. However, these works have sample complexity at least linearly depends on the number of states, which is necessary without additional assumptions [Jaksch et al., 2010].

Various exploration algorithms are proposed for $Q$-learning with function approximation [Azizzadenesheli et al., 2018, Fortunato et al., 2018, Lipton et al., 2018, Osband et al., 2016, Pazis and Parr, 2013]. However, none of these algorithms have polynomial sample complexity guarantees. Li et al. [2011] propose a $Q$-learning algorithm which requires the Know-What-It-Knows oracle. However, it is unknown how to implement such oracle in general. Wen and Van Roy [2013] propose an algorithm for $Q$-learning with function approximation in deterministic systems which works for a family of function classes in the exact setting. For the agnostic setting, the algorithm in [Wen and Van Roy, 2013] can only be applied to a special case called "state aggregation case". See Section 4.3 in [Wen and Van Roy, 2013] for more details. Indeed, as stated in the conclusion of [Wen and Van Roy, 2013], designing provably efficient algorithm for agnostic $Q$-learning with general function approximation is a challenging open problem.

Du et al. [2019] propose an algorithm for $Q$-learning with linear function approximation in the exact setting. The algorithm in [Du et al., 2019] further requires conditions on the optimality gap $\rho$ and a low-variance condition on the transition. Our algorithms also requires conditions on the optimality gap $\rho$ and shares similar recursion-based structures as the algorithm in [Du et al., 2019]. However, our algorithm handles general function classes with bounded Eluder dimension and with approximation error, neither of which can be handled by the algorithm in [Du et al., 2019].

Recently, Du et al. [2020] proved lower bounds for $Q$-learning algorithm in the agnostic setting. As mentioned in the introduction, our algorithm complements the lower bounds in [Du et al., 2020] and demonstrates the tightness of their lower bound. Lattimore and Szepesvari [2019], Van Roy and Dong [2019] also give algorithms in the agnostic setting to demonstrate the tightness of the lower bound in [Du et al., 2020] from other perspectives. Technically, our results are different from those in [Lattimore and Szepesvari, 2019, Van Roy and Dong, 2019] in the following ways. First, we study the standard reinforcement learning setting, where Van Roy and Dong [2019] focus on the

bandit setting and Lattimore and Szepesvari [2019] study both the bandit setting and reinforcement learning with a generative model. Second, for the reinforcement learning result in [Lattimore and Szepesvari, 2019], it is further assumed that $Q$-functions induced by *all* polices can be approximated by linear functions, while in this paper we only assume the optimal $Q$-function can be approximated by a function class with bounded Eluder dimension, which is much weaker than the assumption in [Lattimore and Szepesvari, 2019]. In conjunction with the lower bound in [Du et al., 2020], we give a tight condition $\delta = \widetilde{\Theta}\left(\rho/\sqrt{\dim_E}\right)$ under which there is an algorithm with polynomial sample complexity to find the optimal policy.

Recently, a line of work study $Q$-learning in the linear MDP setting [Yang and Wang, 2019b,a, Jin et al., 2019, Wang et al., 2019]. In the linear MDP setting, it is assumed that both the reward function and the transition operator is linear, which is stronger than the assumption that the optimal $Q$-function is linear studied in this paper. For the linear MDP setting, algorithms with polynomial sample complexity are known, and these algorithms can usually handle approximation errors on the reward function and the transition operator.

# 3  Preliminaries

**Notations** We write $[n]$ to denote the set $\{1, 2, \ldots, n\}$. We use $\|\cdot\|_p$ to denote the $\ell_p$ norm of a vector. For any finite set $S$, we write $\triangle(S)$ to denote the probability simplex.

## 3.1  Episodic Reinforcement Learning

In this paper, we consider Markov Decision Processes with deterministic transition and stochastic reward. Formally, let $\mathcal{M} = (\mathcal{S}, \mathcal{A}, H, P, R)$ be a Markov Decision Process (MDP) where $\mathcal{S}$ is the state space, $\mathcal{A}$ is the action space, $H \in \mathbb{Z}_+$ is the planning horizon, $P : \mathcal{S} \times \mathcal{A} \to \mathcal{S}$ is the deterministic transition function which takes a state-action pair and returns a state, and $R : \mathcal{S} \times \mathcal{A} \to \triangle(\mathbb{R})$ is the reward distribution. When the reward is deterministic, we may regard $R : \mathcal{S} \times \mathcal{A} \to \mathbb{R}$ as a function instead of a distribution. We assume there is a fixed initial state $s_1$.

A policy $\pi : \mathcal{S} \to \triangle(\mathcal{A})$ prescribes a distribution over actions for each state. The policy $\pi$ induces a (random) trajectory $s_1, a_1, r_1, s_2, a_2, r_2, \ldots, s_H, a_H, r_H$ where $a_1 \sim \pi(s_1)$, $r_1 \sim R(s_1, a_1)$, $s_2 = P(s_1, a_1)$, $a_2 \sim \pi(s_2)$, etc. To streamline our analysis, for each $h \in [H]$, we use $\mathcal{S}_h \subseteq \mathcal{S}$ to denote the set of states at level $h$, and we assume $\mathcal{S}_h$ do not intersect with each other.[5] We also assume $\sum_{h=1}^{H} r_h \in [0, 1]$. Our goal is to find a policy $\pi$ that maximizes the expected total reward $\mathbb{E}\left[\sum_{h=1}^{H} r_h \mid \pi\right]$. We use $\pi^*$ to denote the optimal policy.

## 3.2  $Q$-function, $V$-function and the Optimality Gap

An important concept in RL is the $Q$-function. Given a policy $\pi$, a level $h \in [H]$ and a state-action pair $(s, a) \in \mathcal{S}_h \times \mathcal{A}$, the $Q$-function is defined as $Q_h^\pi(s, a) = \mathbb{E}\left[\sum_{h'=h}^{H} r_{h'} \mid s_h = s, a_h = a, \pi\right]$. For simplicity, we denote $Q_h^*(s, a) = Q_h^{\pi^*}(s, a)$. It will also be useful to define the value function of a given state $s \in \mathcal{S}_h$ as $V_h^\pi(s) = \mathbb{E}\left[\sum_{h'=h}^{H} r_{h'} \mid s_h = s, \pi\right]$. For simplicity, we denote $V_h^*(s) = V_h^{\pi^*}(s)$. Throughout the paper, for the $Q$-function $Q_h^\pi$ and $Q_h^*$ and the value function $V_h^\pi$ and $V_h^*$, we may omit $h$ from the subscript when it is clear from the context.

In addition to these definitions, we list below an important concept, the optimality gap, which is widely used in reinforcement learning and bandit literature.

**Definition 3.1** (Optimality Gap). *The optimality gap $\rho$ is defined as $\rho = \inf_{Q^*(s,a) \neq V^*(s)} V^*(s) - Q^*(s, a)$.*

In words, $\rho$ is the smallest reward-to-go difference between the best set of actions and the rest. In this paper we need $\rho$ to be strictly positive. We remark that this is not a restrictive assumption. This assumption is widely used in bandit problems Abbasi-Yadkori et al. [2011], Dani et al. [2008],

Lattimore and Szepesvári [2018]. Recently, Du et al. [2019] gave a provably efficient $Q$-learning algorithm based on this assumption, Simchowitz and Jamieson [2019] showed that with this condition, the agent only incurs logarithmic regret in the tabular setting and Zanette et al. [2019] showed that under this condition, one can remove all horizon dependencies in the sample complexity. Empirically, arguably all environments with a finite action set satisfy the optimality gap conditions. In Atari-games, e.g., Freeway, the optimal $Q$ value is often distinctive from the rest of actions. For board games, e.g. tic-tac-toe, Chess, etc, most states have zero rewards except for the winning states. Hence, every optimal action has a $Q$-value of 1 and the rest actions have a $Q$-value of 0, in which case $\rho = 1$ by Definition 3.1.

### 3.3 Function Approximation and Eluder Dimension

When the state space is large, we need structures on the state space so that reinforcement learning methods can generalize. For a given function class $\mathcal{F}$, each $f \in \mathcal{F}$ is a function that maps a state-action pair to a real number. For a given MDP and a function class $\mathcal{F}$, we define the approximation error to the optimal $Q$-function as follow.

**Definition 3.2** (Approximation Error). *For a given MDP and a function class $\mathcal{F}$, the approximation error $\delta$ is defined to be $\delta = \inf_{f \in \mathcal{F}} \sup_{(s,a) \in \mathcal{S} \times \mathcal{A}} |f(s,a) - Q^*(s,a)|$.*

Here, the approximation error $\delta$ characterizes how well the given function class $\mathcal{F}$ approximates the optimal $Q$-function. When $\delta = 0$, then optimal $Q$-function can be perfectly predicted by the function class, which has been studied in previous papers [Wen and Van Roy, 2013, Du et al., 2019]. In this paper, we focus the case $\delta > 0$.

An important function class is the class of linear functions. We assume the agent is given a feature extractor $\phi : \mathcal{S} \times \mathcal{A} \to \mathbb{R}^d$ where $\|\phi(s,a)\|_2 \leq 1$ for all state-action pairs. The feature extractor can be hand-crafted or a pre-trained neural network that transforms a state-action pair to a $d$-dimensional embedding. Given the feature extractor $\phi$, we define the class of linear functions as follow.

**Definition 3.3.** *For a vector $\theta \in \mathbb{R}^d$, we define $f_\theta(s,a) = \theta^\top \phi(s,a)$. The class of linear functions is defined as $\mathcal{F} = \{f_\theta \mid \|\theta\|_2 \leq 1\}$.*

Here we assume $\|\theta\|_2 \leq 1$ only for normalization purposes.

For general function classes, an important concept is the *Eluder dimension*, for which we first need to introduce the concept of $\epsilon$-dependence.

**Definition 3.4** ($\epsilon$-dependence [Russo and Van Roy, 2013]). *For a function class $\mathcal{F}$, we say a state-action pair $(s,a)$ is $\epsilon$-dependent on state-action pairs $\{(s_1,a_1),\ldots,(s_n,a_n)\} \subset \mathcal{S} \times \mathcal{A}$ with respect to $\mathcal{F}$ if for all $f_1, f_2 \in \mathcal{F}$,*

$$\sum_{i=1}^{n} |f_1(s_i,a_i) - f_2(s_i,a_i)|^2 \leq \epsilon^2 \implies |f_1(s,a) - f_2(s,a)|^2 \leq \epsilon^2.$$

*Further, $(s,a)$ is $\epsilon$-independent of state-action pairs $\{(s_1,a_1),\ldots,(s_n,a_n)\}$ if $(s,a)$ is not $\epsilon$-dependent on state-action pairs $\{(s_1,a_1),\ldots,(s_n,a_n)\}$.*

Now, we recall the definition of $\epsilon$-Eluder dimension as introduced in Russo and Van Roy [2013].

**Definition 3.5** ($\varepsilon$-Eluder Dimension). *For a function class $\mathcal{F}$, the $\epsilon$-Eluder dimension $\dim_E(\mathcal{F}, \epsilon)$ is the length of the longest sequence of elements in $\mathcal{S} \times \mathcal{A}$ such that every element is $\epsilon'$-independent of its predecessors for some $\epsilon' \geq \epsilon$.*

As an example, when $\mathcal{F}$ is the class of linear functions with norm $\|\theta\|_2 \leq 1$ and $\|\phi(s,a)\|_2 \leq 1$, the $\varepsilon$-Eluder dimension $\dim_E(\mathcal{F}, \epsilon)$ is $O(d \log(1/\epsilon))$ as noted in Example 4 in Russo and Van Roy [2013]. We refer interested readers to Russo and Van Roy [2013] for more examples.

We remark that in this paper, the sample complexity of our algorithm depends on the $\varepsilon$-Eluder dimension introduced in Russo and Van Roy [2013] instead of the the Eluder dimension introduced in Wen and Van Roy [2013], since the Eluder dimension introduced in Wen and Van Roy [2013] is defined for the exact case and therefore cannot handle approximation errors.

---

**Algorithm 1** Main Algorithm

---

1: Initialize the current policy $\pi$ arbitrarily
2: **set** $C = \rho^2/16 \cdot I \in \mathbb{R}^{d \times d}$
3: **set** $Y = 0 \in \mathbb{R}^d$
4: **invoke** Explore($s_1$)
5: **return** $\pi$

---

---

**Algorithm 2** Explore($s$)

---

1: **for** $a \in \mathcal{A}$ **do**
2:    **if** $\phi(s,a)^\top C^{-1} \phi(s,a) \leq 1$ **then**
3:       **set** $\hat{Q}(s,a) = \phi(s,a)^\top C^{-1} Y$
4:    **else**
5:       **let** $s' = P(s,a)$
6:       **set**

$$\hat{Q}(s,a) = \begin{cases} r(s,a) & \text{if } s \in \mathcal{S}_H \\ \text{Explore}(s') + r(s,a) & \text{otherwise} \end{cases}$$

7:       **set** $C = C + \phi(s,a)\phi(s,a)^\top$, $Y = Y + \phi(s,a)\hat{Q}(s,a)$
8:    **end if**
9: **end for**
10: **set** $\pi(s) = \operatorname{argmax}_{a \in \mathcal{A}} \hat{Q}(s,a)$.
11: **return** $r(s, \pi(s)) + \text{Explore}(P(s, \pi(s)))$

---

## 4 Algorithm for Linear Functions

In this section, we consider the special case where $\mathcal{F}$ is the class of linear functions to demonstrate the high-level approach of our algorithm and the intuition behind the analysis. For simplicity, we also assume that the size of action space $\mathcal{A}$ is bounded by a constant and the reward is deterministic. We show how to remove these assumptions in the following sections.

Our goal is to show when $\rho = \Omega(\delta\sqrt{d \log(1/\rho)})$, Algorithm 1 learns the optimal policy $\pi^*$ using nearly linear number of trajectories.

**Theorem 4.1.** *Suppose* $\rho \geq 4\delta(\sqrt{2d \log(16/\rho^2)} + 1)$. *Algorithm 1 returns the optimal policy* $\pi^*$ *using at most* $O(d \log(1/\rho))$ *trajectories.*

The complete proof is provided in the supplementary material. On a high level, our algorithm is divided into two parts: Algorithm 1 in which we define the main loop and Algorithm 2 in which we define a recursion-based subroutine Explore($s$) to calculate the optimal values. Intuitively, the subroutine Explore($s$) should return $V^*(s)$, and upon the termination of Explore($s$) we should have $\pi(s) = \pi^*(s)$.

In our algorithm, we maintain a dataset to store the features of a subset of the state-action pairs $\phi(s,a)$ and their optimal $Q$-values $Q^*(s,a)$. Here, the matrix $C \in \mathbb{R}^d$ is the covariance of the dataset, i.e., $C = \sum \phi(s,a)\phi(s,a)^\top$ and $Y = \sum \phi(s,a)Q^*(s,a)$. In order to predict the optimal $Q$-value of an unseen state-action pair $(s,a)$ using least squares, we may directly calculate $\phi(s,a)^\top C^{-1} Y$ if $C$ is invertible. We use a ridge term of $\rho^2/16$ to make sure $C$ is always invertible.

The high-level idea behind our algorithm is simple: we use least squares to predict the optimal $Q$-value whenever possible, and use recursions to figure out the optimal $Q$-value otherwise. One technical subtlety here is: What condition should we check to decide whether we can calculate the optimal $Q$-value directly by least squares or we need to make recursive calls? This condition needs to be chosen carefully, since if we make too many recursive calls, the overall sample complexity will be unbounded, and if we make too few recursive calls, the optimal $Q$-values estimated by linear squares will be inaccurate which affects the correctness of the algorithm.

In Line 2 of Explore($s$), we check whether $\phi(s,a)^\top C^{-1} \phi(s,a) \leq 1$, which is the condition we use to decide whether we should make recursive calls or calculate the optimal $Q$-value directly by least squares. Here $\phi(s,a)^\top C^{-1} \phi(s,a)$ is the variance of the prediction, which is common in UCB-type algorithm for linear contextual bandit (see e.g. Li et al. [2010]). In our algorithm, instead of using

$\phi(s, a)^\top C^{-1} \phi(s, a)$ as an uncertainty bonus, we directly check its magnitude to decide whether the linear predictor learned on the collected dataset generalizes well on the new data $\phi(s, a)$ or not. The effectiveness of such a choice follows from the following lemma which bounds the number of recursive calls made by our algorithm.

**Lemma 4.2.** *Line 7 is executed for at most $2d \log(16/\rho^2)$ times.*

A proof is provided in the supplementary material. Moreover, in order to make sure that the value returned by Explore$(s)$ is accurate, in Line 11 of Explore$(s)$, we make recursive calls instead of using the estimated $Q$-values $\hat{Q}$. As will be shown in the supplementary material, such a choice guarantees that the value returned by Explore$(s)$ always equals $V^*(s)$.

Lastly, we want to remark that if we use a $\rho' < \rho$ in the algorithm, the algorithm is still correct and the sample complexity will be $O(d \log(1/\rho'))$. For unknown $\rho$, one can use an exponential search in a suitable range which will only increase the sample complexity by a logarithmic factor.

## 5  General Result

In this section, we consider the general case where $\mathcal{F}$ is an arbitrary function class and provide a provably efficient algorithm which is a generalization of the algorithm in Section 4. Note that we make no assumptions on the action space $\mathcal{A}$. For simplicity, we assume that the reward is deterministic. We show how to remove this assumption in the supplementary material. We first define the Maximum Uncertainty Oracle which allows us to work with arbitrary action space.

### 5.1  Maximum Uncertainty Oracle

As discussed the previous section, it is useful to identify actions for which we can not accurately compute the optimal $Q$-value using the least-squares predictor. We formalize this intuition to arrive at the following oracle which finds the action with largest "uncertainty" for a given state $s$. We note that similar oracles were also used in [Du et al., 2019].

**Definition 5.1** (Oracle$(s, \delta, Y)$)**.** *Given a state $s \in \mathcal{S}$, $\delta \geq 0$ and a set of state-action pairs $Y \subseteq \mathcal{S} \times \mathcal{A}$, define*

$$(\hat{a}, \hat{f}_1, \hat{f}_2) = \underset{a \in A, f_1, f_2 \in \mathcal{F}}{\operatorname{argmax}} |f_1(s, a) - f_2(s, a)|^2 \tag{1}$$

$$s.t. \quad \frac{1}{|Y|} \sum_{(s', a') \in Y} |f_1(s', a') - f_2(s', a')|^2 \leq \delta^2. \tag{2}$$

*The oracle returns $(\hat{a}, |\hat{f}_1(s, \hat{a}) - \hat{f}_2(s, \hat{a})|^2)$.*

To motivate this oracle, suppose $f_2$ is the function that gives the best approximation of the optimal $Q$-function, i.e., the optimizer $f$ in Definition 3.2. In this scenario, we know $f_1$ predicts well on state-action pairs $(s', a') \in Y$ which is implied by the constraint. Note that since we maximize over the entire function class $\mathcal{F}$, $\hat{a}$ is the action with largest uncertainty. If $|\hat{f}_1(s, \hat{a}) - \hat{f}_2(s, \hat{a})|^2$ is small, then we can predict well on state $s$ for all actions. Otherwise, if we cannot predict well on state $s$ for some action, so we need to explore and return the action with largest uncertainty.

**Remark 1.** *When $\mathcal{F}$ is the class of linear functions, evaluating the oracle's response amounts to solving:*

$$(\hat{a}, \hat{\theta}_1, \hat{\theta}_2) = \underset{a \in A, \theta_1, \theta_2 \in \mathcal{F}}{\operatorname{argmax}} |(\theta_1 - \theta_2)^\top \phi(s, a)|^2$$

$$s.t. \quad (\theta_1 - \theta_2)^\top \left( \frac{1}{|Y|} \sum_{(s', a') \in Y} \phi(s', a') \phi(s', a')^\top \right) (\theta_1 - \theta_2) \leq \delta^2.$$

*In this case, using the notation in the algorithm in Section 4, it can be seen that the oracle returns the action $a \in \mathcal{A}$ which maximizes $\phi(s, a)^\top C^{-1} \phi(s, a)$.*

---

**Algorithm 3** Main Algorithm

---
1: Initialize the current policy $\pi$ and $f$ arbitrarily.
2: **set** $Y = \{\}$
3: **invoke** Explore($s_1$)
4: **return** $\pi$

---

---

**Algorithm 4** Explore($s$)

---
1: **set** $(a, r) = \text{Oracle}(s, 2\delta, Y)$
2: **while** $r > |\frac{\rho}{2} - \delta|$ **do**
3:     **set** $Y = Y \cup \{(s, a, \text{Explore}(P(s, a)) + r(s, a))\}$
4:     **set** $(a, r) = \text{Oracle}(s, 2\delta, Y)$
5: **end while**
6: **set** $f = \text{argmin}_{f \in \mathcal{F}} \sum_{(s_i, a_i, y_i) \in Y} |f(s_i, a_i) - y_i|^2$
7: **set** $\pi(s) = \text{argmax}_{a \in \mathcal{A}} f(s, a)$
8: **return** $r(s, \pi(s)) + \text{Explore}(P(s, \pi(s)))$

---

## 5.2 Algorithm

In this section, we present the high level intuition for the Algorithm 3. Our goal is to show that when $\rho = \Omega(\delta \sqrt{\dim_E(\mathcal{F}, \rho)})$, our algorithm learns the optimal policy $\pi^*$ using linear number of trajectories (in terms of Eluder dimension).

**Theorem 5.1.** *Suppose*

$$\rho \geq 6\sqrt{2}\delta \sqrt{\dim_E(\mathcal{F}, \frac{\rho}{4})}. \tag{3}$$

*Then Algorithm 3 returns the optimal policy $\pi^*$ using at most $O(\dim_E(\mathcal{F}, \rho/4))$ trajectories.*

The complete proof is provided in the supplementary material. Similar to the algorithm for linear functions given in Section 4, the algorithm for general function class is divided into two parts: Algorithm 3 and a subroutine Explore($s$). Intuitively, the subroutine Explore(s) should return $V^*(s)$, and upon the termination of Explore($s$), we should have $\pi(s) = \pi^*(s)$.

In our algorithm, we maintain a dataset to store the state-action pairs $(s, a)$ and their optimal $Q$-values $Q^*(s, a)$. In order to predict the optimal $Q$-value of an unseen state-action pair $(s, a)$, we find the best predictor on the dataset using least squares, and use it to predict on $(s, a)$.

Similar to the algorithm in Section 4, the high level idea is that we use least squares to predict the optimal $Q$-value whenever possible, and otherwise we explore the environment. In Line 2, we check for a state $s$, whether the Maximum Uncertainty Oracle reports an uncertainty $r > |\rho/2 - \delta|$. Such a choice guarantees that the value returned by Explore($s$) always equals $V^*(s)$ and also, as we prove, upper bounds the number of times we explore, i.e., execute Line 3, by the Eluder dimension of function class $\mathcal{F}$.

**Lemma 5.2.** *For any constant $c > 1$, suppose $\rho \geq 4\delta \sqrt{\frac{c \dim_E(F, \frac{\rho}{4}) - 1}{c - 1}} + 2\delta$. Then we have $|Y| \leq c \dim_E(F, \frac{\rho}{4})$.*

A proof is provided in the supplementary material. The proof relies on definition of the Eluder dimension and the Maximum Uncertainty Oracle. We remark that when applied to linear functions, using the notation in the algorithm in Section 4, the subroutine Explore(s) keeps finding an action $a \in \mathcal{A}$ which maximizes $\phi(s, a)^\top C^{-1} \phi(s, a)$ (see Remark 1) until $\phi(s, a)^\top C^{-1} \phi(s, a)$ is below a threshold for all actions $a \in \mathcal{A}$. Therefore, our algorithm is a generalization of the algorithm in Section 4.

Again, we remark that while Algorithm 3 depends on both $\rho$ and $\delta$, one can use an exponential search for $\rho$ and $\delta$, and the sample complexity will increase mildly.

# 6 Conclusion

In this paper, we propose a novel provably efficient recursion-based algorithm for agnostic $Q$-learning with general function approximation in deterministic systems. We obtain a sharp characterization on the relation between the approximation error and the optimality gap, and also a tight sample complexity. We help address the open problem raised by Wen and Van Roy [2013].

## Broader Impact

The focus of this paper is purely theoretical, and thus a broader impact discussion is not applicable.

## Footnotes

[5]This assumption is only for the sake of presentation. Our result can be easily generalized to the case when this assumption does not hold.

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
