[Supplementary Material]

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

## A  Proof of Proposition 1.2

In this section, we briefly discuss how to generalize the results in [Du et al., 2020] to prove Proposition 1.2. We first recall Theorem 4.1 in [Du et al., 2020].

**Proposition A.1** (Theorem 4.1 in [Du et al., 2020]). *There exists a family of deterministic systems $\mathcal{M}$ such that for any $M \in \mathcal{M}$, the following conditions hold. There exists a feature extractor $\phi : \mathcal{S} \times \mathcal{A} \to \mathbb{R}^d$ and $\theta_1, \theta_2, \ldots, \theta_H \in \mathbb{R}^d$ such that $d = O(H/\delta^2)$, and for any $h \in [H]$ and any $(s, a) \in \mathcal{S}_h \times \mathcal{A}$,*

$$|Q^*(s, a) - \theta_h^\top \phi(s, a)| \le \delta.$$

*Moreover, for the deterministic systems in $\mathcal{M}$, any algorithm that returns a $1/2$-optimal policy with probability $0.9$ needs to sample $\Omega(2^H)$ trajectories.*

We first note that the assumption in Proposition A.1 is slightly different from ours. In this paper, we assume there exists a single vector $\theta \in \mathbb{R}^d$ such that for any $(s, a) \in \mathcal{S} \times \mathcal{A}$,

$$|Q^*(s, a) - \theta^\top \phi(s, a)| \le \delta.$$

However, the lower bound in [Du et al., 2020] can still be generalized to hold under our assumption, if one breaks the feature space into $H$ blocks so that each block contains $d/H$ coordinates, and for any state $s_1 \in \mathcal{S}_1$ and $a \in \mathcal{A}$, $\phi(s_1, a)$ contains non-zero entries only in the first block, and for any state $s_2 \in \mathcal{S}_2$ and $a \in \mathcal{A}$, $\phi(s_2, a)$ contains non-zero entries only in the second block, etc. By doing so, we need to change the condition $d = O(H/\delta^2)$ to $d = O(H^2/\delta^2)$.

Moreover, in order to prove an $\Omega(2^C)$ sample complexity lower bound, one only needs to use the first $C$ levels in the family of deterministic systems in Proposition A.1, and add $H - C$ dummy levels so that there are $H$ levels in total. In this case, Proposition A.1 requires $d = O(C^2/\delta^2)$, or equivalently, $\delta = \Omega(C/\sqrt{d})$.

Finally, by scrutinizing the construction in [Du et al., 2020], it can be seen that the optimality gap $\rho = 1$. In general, for a given value $\rho \le 1$, we can scale all reward values and the vector $\theta$ in the original construction by $\rho$. By doing so, the approximation error $\delta = \Omega(C/\sqrt{d} \cdot \rho)$.

## B  Missing Proofs in Section 4

*Proof of Theorem 4.1.* Recall that by Definition 3.2 and Definition 3.3, there exists $\theta \in \mathbb{R}^d$ with $\|\theta\|_2 \le 1$ such that $|Q^*(s, a) - \theta^\top \phi(s, a)| \le \delta$ for all state-action pairs $(s, a)$.

Since the sample complexity of our algorithm equals the number of times we execute Line 5 in Explore($s$), following Lemma 4.2, the sample complexity of our algorithm is $O(d \log(1/\rho))$.

To complete the proof, it is sufficient to prove the following induction hypothesis for all levels $h \in [H]$.

**Induction Hypothesis.**

&emsp; I  When Line 6 is executed for any state $s \in \mathcal{S}_h$, $\hat{Q}(s, a) = Q^*(s, a)$.

&emsp; II  Each time Line 10 in Explore($s$) is executed for any state $s \in \mathcal{S}_h$, we have $\pi(s) = \pi^*(s)$, and the value returned by Explore($s$) equals $V^*(s)$.

For the above induction hypothesis, the base case $h = H$ is clearly true. Now we assume the induction hypothesis holds for all levels $H, \ldots, h + 1$ and prove it holds for all levels $h \in [H]$.

**Induction Hypothesis I.**  This follows from Induction Hypothesis II for level $h+1$ and the Bellman equations.

**Induction Hypothesis II.**  By Induction Hypothesis I and Definition 3.1, we only need to show when Line 3 is executed, we have $|\hat{Q}(s, a) - Q^*(s, a)| \le \rho/2$, in which case we have $\pi(s) = \pi^*(s)$. To verify this, note that

$$|\phi(s, a)^\top C^{-1} Y - Q^*(s, a)| \le |\phi(s, a)^\top C^{-1} Y - \theta^\top \phi(s, a)| + |Q^*(s, a) - \theta^\top \phi(s, a)|.$$

The second term is bounded by $\delta$. For the first term, we write $\Phi$ to be a matrix whose $i$-th column is the $i$-th $\phi(s,a)$ vector in the summation. Recall that

$$C = \left(\sum \phi(s,a)\phi(s,a)^\top\right) + \rho^2/16 \cdot I = \Phi\Phi^\top + \rho^2/16 \cdot I$$

and

$$Y = \sum \phi(s,a)Q^*(s,a)$$

by Induction Hypothesis I. Moreover,

$$Y = \sum \phi(s,a)(\phi(s,a)^\top\theta + b(s,a))$$

where $|b(\cdot,\cdot)| \le \delta$. Thus, the first term can be upper bounded by

$$\|\phi(s,a)^\top C^{-1}\Phi\|_1 \cdot \delta + \left|\phi(s,a)^\top(C^{-1}\Phi\Phi^\top - I)\theta\right|.$$

For the first term, by Lemma 4.2 there are at most $2d\log(16/\rho^2)$ columns in $\Phi$. When Line 3 is executed, we must have $\phi(s,a)^\top C^{-1}\phi(s,a) \le 1$. Using Lemma B.1 we have

$$
\begin{aligned}
&\|\phi(s,a)^\top C^{-1}\Phi\|_1 \\
\le{}& \sqrt{2d\log(16/\rho^2)} \cdot \|\phi(s,a)^\top C^{-1}\Phi\|_2 \\
={}& \sqrt{2d\log(16/\rho^2)} \cdot \sqrt{\phi(s,a)^\top C^{-1}\Phi\Phi^\top C^{-1}\phi(s,a)} \\
\le{}& \sqrt{2d\log(16/\rho^2)}.
\end{aligned}
$$

For the second term, since $\|\theta\|_2 \le 1$ and $\phi(s,a)^\top C^{-1}\phi(s,a) \le 1$, by Cauchy-Schwarz and Lemma B.1, we have

$$|\phi(s,a)^\top(C^{-1}\Phi^\top\Phi - I)\theta| \le \|\phi(s,a)^\top(C^{-1}\Phi^\top\Phi - I)\|_2 \le \rho/4.$$

All together we get

$$|\phi(s,a)^\top C^{-1}Y - Q^*(s,a)| \le \rho/2$$

which completes the proof. $\qquad\square$

*Proof of Lemma 4.2.* Suppose Line 7 has been executed for $T$ times, since $\|\phi(s,a)\|_2 \le 1$, the trace of $\phi(s,a)\phi(s,a)^\top$ is upper bounded by $\|\phi(s,a)\|_2^2 \le 1$. By additivity of trace, the trace of $C$ is upper bounded by

$$T + d \cdot \rho^2/16$$

since initially the trace of $C$ is $d \cdot \rho^2/16$. By AM-GM,

$$\det(C) \le (T/d + \rho^2/16)^d.$$

However, each time Line 7 is executed, by matrix determinant lemma, $\det(C)$ will be increased by a factor of

$$1 + \phi(s_h,a)^\top C^{-1}\phi(s_h,a) \ge 2.$$

Moreover, initially $\det(C) = (\rho^2/16)^d$. Thus,

$$2^T(\rho^2/16)^d \le (T/d + \rho^2/16)^d,$$

which proves the lemma. $\qquad\square$

**Lemma B.1.** *For any positive semi-definite $M \in \mathbb{R}^{d\times d}$, $\alpha > 0$ and $x \in \mathbb{R}^d$ such that $x^\top(M + \alpha \cdot I)^{-1}x \le 1$, we have*

- $\|(M(M + \alpha \cdot I)^{-1} - I)x\|_2 \le \alpha$;

- $x^\top(M + \alpha \cdot I)^{-1}M(M + \alpha \cdot I)^{-1}x \le 1$.

*Proof.* We use $M = U^T \Lambda U$ to denote the spectral decomposition of $M$, where $\Lambda$ is a diagonal matrix with non-negative entries. We use $\Lambda_i$ to denote the $i$-th diagonal entry of $\Lambda$ and let $y = Ux$. By the assumption, it holds that

$$\sum_{i=1}^{d} \frac{y_i^2}{\Lambda_i + \alpha} \leq 1.$$

Clearly,

$$\|(M(M + \alpha \cdot I)^{-1} - I)x\|_2^2$$
$$= \sum_{i=1}^{d} y_i^2 \cdot \left( \frac{\Lambda_i}{\Lambda_i + \alpha} - 1 \right)^2 = \sum_{i=1}^{d} y_i^2 \cdot \left( \frac{\alpha}{\Lambda_i + \alpha} \right)^2 \leq \alpha$$

and

$$x^\top (M + \alpha \cdot I)^{-1} M (M + \alpha \cdot I)^{-1} x$$
$$= \sum_{i=1}^{d} y_i^2 \cdot \frac{\Lambda_i}{(\Lambda_i + \alpha \cdot I)^2} \leq 1.$$

$\square$

# C   Missing Proofs in Section 5

*Proof of Theorem 5.1.* Firstly, using Lemma 5.2 with $c = 18$ we have

$$|Y| \leq 18 \dim_E(\mathcal{F}, \frac{\rho}{4}), \tag{4}$$

i.e. Line 3 is executed for at most $18 \dim_E(F, \rho/4)$ times and therefore the sample complexity of our algorithm is $O(\dim_E(\mathcal{F}, \rho/4))$.

To complete the proof, it is sufficient to prove the following induction hypothesis for all levels $h \in [H]$.

**Induction Hypothesis.**

I For any state $s \in \mathcal{S}_h$, when Line 6 in Explore$(s)$ is executed, we have $y_i = Q^*(s_i, a_i)$ for all $(s_i, a_i, y_i) \in Y$.

II For any state $s \in \mathcal{S}_h$, when Line 7 in Explore$(s)$ is executed, we have $\pi(s) = \pi^*(s)$, and the value returned by Explore$(s)$ is $V^*(s)$.

For the above induction hypothesis, the base case $h = H$ is clearly true. Now we assume the induction hypothesis holds for all levels $H, \ldots, h+1$ and prove it holds for all levels $h \in [H]$.

**Induction Hypothesis I.**   From Induction Hypothesis II for level $h+1$, it follows that value returned by Explore$(P(s, a))$ is $V^*(P(s, a))$ for all $a \in \mathcal{A}$. Then, Induction Hypothesis I follows from the Bellman equations.

**Induction Hypothesis II.**   It suffices to show that for any state $s \in \mathcal{S}_h$, when Line 7 in Explore$(s)$ is executed, for all actions $a \in \mathcal{A}$

$$|f(s, a) - Q^*(s, a)| \leq \frac{\rho}{2}. \tag{5}$$

First, there exists $f^* \in \mathcal{F}$ such that for all $(s_i, a_i, y_i) \in Y$,

$$|f^*(s_i, a_i) - Q^*(s_i, a_i)| \leq \delta. \tag{6}$$

From Induction Hypothesis I, for all $(s_i, a_i, y_i) \in Y$

$$y_i = Q^*(s_i, a_i). \tag{7}$$

From Equation (6) and (7), it follows that

$$\sum_{(s_i, a_i, y_i) \in Y} |f^*(s_i, a_i) - y_i|^2 \le |Y|\delta^2. \tag{8}$$

Since, we execute Line 6 and $f^* \in \mathcal{F}$, from Equation (8), it follows that

$$\sum_{(s_i, a_i, y_i) \in Y} |f(s_i, a_i) - y_i|^2 \le |Y|\delta^2. \tag{9}$$

We split the analysis into two cases:

(1)  we consider actions for which we execute Line 3 and

(2)  we consider rest of the actions.

**Case 1:**  We now prove Equation (5) for all actions $a$ for which we execute Line 3. Using Equation (4), (7) and (9), we get that for actions $a$ for which we executed Line 3 (since then we added it to $Y$)

$$|f(s, a) - Q^*(s, a)| \le \sqrt{18 \dim_E(F, \frac{\rho}{4})} \delta \le \frac{\rho}{2} \tag{10}$$

where the last step follows from our assumption on $\rho$ (Equation (3)).

**Case 2:**  We now prove this for rest of the actions $a$. From Equation (6), (7), (9) and triangle inequality for the $\ell_2$ norm, we get

$$\sum_{(s_i, a_i, y_i) \in Y} |f^*(s_i, a_i) - f(s_i, a_i)|^2 \le 4|Y|\delta^2. \tag{11}$$

Also, since we did not add this action to $Y$, by the definition of the oracle (Definition 5.1), we get

$$|f^*(s, a) - f(s, a)| \le \frac{\rho}{2} - \delta. \tag{12}$$

Therefore,

$$|Q^*(s, a) - f(s, a)| \le \frac{\rho}{2} \tag{13}$$

which completes the proof. $\qquad\square$

*Proof of Lemma 5.2.*  For some $n > 0$, assume

$$Y = \{(s_1, a_1, y_1), \dots, (s_n, a_n, y_n)\}.$$

We will show that $n$ is upper bounded by Eluder dimension. When we add $(s_j, a_j, y_j)$ to $Y$ at Line 3,

1. The condition at Line 2 must be True i.e. from Equation (1), there exists $f_1, f_2 \in F$ such that $|f_1(s_j, a_j) - f_2(s_j, a_j)| > \frac{\rho}{2} - \delta$.

2. Observe that for any subsequence $B \subset \{(s_1, a_1), \dots, (s_{j-1}, a_{j-1})\}$ where $(s_j, a_j)$ is $(\frac{\rho}{2} - \delta)$-dependent on $B$ (Definition 3.4),

$$\sum_{(s, a) \in B} |f_1(s, a) - f_2(s, a)|^2 \ge (\frac{\rho}{2} - \delta)^2. \tag{14}$$

3. Therefore, if there are $K$ disjoint subsequences in $\{(s_1, a_1), \dots, (s_{j-1}, a_{j-1})\}$ such that $(s_j, a_j)$ is $(\frac{\rho}{2} - \delta)$-dependent on all of them, then

$$\sum_{i=1}^{j-1} |f_1(s_i, a_i) - f_2(s_i, a_i)|^2 \ge K(\frac{\rho}{2} - \delta)^2. \tag{15}$$

4. However, using Equation 2, we have that

$$\sum_{i=1}^{j-1} |f_1(s_i, a_i) - f_2(s_i, a_i)|^2 \le (j-1)(2\delta)^2. \tag{16}$$

Therefore, we can upper bound for any state-action pair $(s_j, a_j) \in \{(s_1, a_1), \ldots, (s_n, a_n)\}$, the number of disjoint subsequences $K$ in $\{(s_1, a_1), \ldots, (s_{j-1}, a_{j-1})\}$ that $(s_j, a_j)$ is $(\frac{\rho}{2} - \delta)$-dependent on, i.e.

$$K \le \frac{(j-1)(2\delta)^2}{(\frac{\rho}{2} - \delta)^2}.$$

Moreover, it follows from the proof of Proposition 3 in [Russo and Van Roy, 2013] that for any sequence of state-action pairs say $\{(s_1, a_1), \ldots, (s_n, a_n)\}$, there exists a $(s_j, a_j)$ which is $(\frac{\rho}{2} - \delta)$-dependent on at least $\frac{n}{\dim_E(F, \frac{\rho}{2} - \delta)} - 1$ disjoint subsequences in $\{(s_1, a_1), \ldots, (s_{j-1}, a_{j-1})\}$. Therefore,

$$\frac{n}{\dim_E(F, \frac{\rho}{2} - \delta)} - 1 \le K \le \frac{(j-1)(2\delta)^2}{(\frac{\rho}{2} - \delta)^2} \tag{17}$$

and thus

$$n \le \dim_E(F, \frac{\rho}{2} - \delta) \left( \frac{(n-1)(2\delta)^2}{(\frac{\rho}{2} - \delta)^2} + 1 \right). \tag{18}$$

As $\rho > 4\delta$, we get

$$n \le \dim_E(F, \frac{\rho}{4}) \left( \frac{(n-1)(2\delta)^2}{(\frac{\rho}{2} - \delta)^2} + 1 \right) \tag{19}$$

which follows from definition of Eluder dimension since $a < b$ implies $\dim_E(F, a) \ge \dim_E(F, b)$. For any $\rho$ and $c > 1$ such that

$$\rho \ge 2 \left( 2\sqrt{\frac{c \dim_E(F, \frac{\rho}{4}) - 1}{c - 1}} + 1 \right) \delta \tag{20}$$

we get from Equation (19) that

$$n \le c \dim_E(F, \frac{\rho}{4}). \tag{21}$$

$\square$

# D   Extension to Stochastic Rewards

---
**Algorithm 5** Main Algorithm

---
1: Initialize the current policy $\pi$ and $f$ arbitrarily
2: **set** $Y = \{\}$
3: **invoke** Explore($s_1$)

---

In this section, we extend our algorithm and analysis to stochastic rewards, i.e., reward $r(s, a) \sim R(s, a)$ is a random variable with expectation $\bar{r}(s, a)$ and $r(s, a) \in [0, 1]$.

## D.1   Algorithm

We modify Explore($s$) such that whenever previously we used $r(s, a)$, we use the empirical mean $\hat{r}(s, a)$ of $n$ samples from $R(s, a)$ to get a good estimate of the expected reward $\bar{r}(s, a)$. For our algorithm, we set

$$n = \frac{H^2}{2\delta_r^2} \log \frac{18 \dim_E(\mathcal{F}, \rho/4)H}{p}, \tag{22}$$

where $\delta_r$ is a parameter to be chosen and $p$ is the failure probability of the algorithm.

---

**Algorithm 6** Explore($s$)

---

1: **set** $(a, r) = \mathsf{Oracle}(s, 2(\delta + \delta_r), Y)$
2: **while** $r > |\frac{\rho}{2} - \delta|$ **do**
3:   **set** $\hat{r}(s, a)$ to be the empirical mean of $n = \frac{H^2}{2\delta_r^2} \log \frac{18 \dim_E(\mathcal{F}, \rho/4) H}{p}$ samples from $R(s, a)$
4:   **set**
$$Y = \begin{cases} Y \cup \{(s, a, \hat{r}(s, a))\} & s \in \mathcal{S}_H \\ Y \cup \{(s, a, \mathsf{Explore}(P(s, a)) + \hat{r}(s, a))\} & \text{otherwise} \end{cases}$$
5:   **set** $(a, r) = \mathsf{Oracle}(s, 2(\delta + \delta_r), Y)$
6: **end while**
7: **set** $f = \operatorname{argmin}_{f \in \mathcal{F}} \sum_{(s_i, a_i, y_i) \in Y} |f(s_i, a_i) - y_i|^2$
8: **set** $\pi(s) = \operatorname{argmax}_{a \in \mathcal{A}} f(s, a)$
9: **return**
$$\begin{cases} \hat{r}(s, \pi(s)) & s \in \mathcal{S}_H \\ \hat{r}(s, \pi(s)) + \mathsf{Explore}(P(s, \pi(s))) & \text{otherwise} \end{cases}$$

---

## D.2   Analysis

**Theorem D.1.** *Suppose*

$$\rho \geq 6\sqrt{2}(\delta + \delta_r)\sqrt{\dim_E(\mathcal{F}, \rho/4)} + 2\delta_r. \tag{23}$$

*Algorithm 5 returns the optimal policy $\pi^*$ with probability $1 - p$.*

**Remark 2.** *Note that by setting*

$$\delta_r = \frac{\rho}{24\sqrt{2 \dim_E(\mathcal{F}, \rho/4)}} \quad \text{and} \quad \rho \geq 12\sqrt{2}\delta\sqrt{\dim_E(\mathcal{F}, \rho/4)}, \tag{24}$$

*Theorem D.1 implies that Algorithm 5 returns the optimal policy $\pi^*$ with probability $1 - p$ using at most*

$$\frac{\mathrm{poly}(\dim_E(\mathcal{F}, \rho/4), H)}{\rho^2} \log(1/p)$$

*trajectories.*

Now we formally prove Theorem D.1.

*Proof of Theorem D.1.* Firstly, for $c = 18$, following Lemma D.2, we have

$$|Y| \leq 18 \dim_E(\mathcal{F}, \rho/4), \tag{25}$$

i.e. Line 4 is executed for at most $18 \dim_E(F, \rho/4)$ times.

To complete the proof, it is sufficient to prove the following induction hypothesis for all levels $h \in [H]$.

**Induction Hypothesis.**

1. For any state $s \in \mathcal{S}_h$, when Line 7 in Explore($s$) is executed, we have

$$y_i \in \left[ Q^*(s_i, a_i) - \frac{H - h + 1}{H}\delta_r, Q^*(s_i, a_i) + \frac{H - h + 1}{H}\delta_r \right]$$

   for all $(s_i, a_i, y_i) \in Y$.

2. For any state $s \in \mathcal{S}_h$, when Line 8 in Explore($s$) is executed, we have $\pi(s) = \pi^*(s)$, and the value returned by Explore($s$) is in

$$\left[ V^*(s) - \frac{H - h + 1}{H}\delta_r, V^*(s) + \frac{H - h + 1}{H}\delta_r \right].$$

Note that the base case $h = H$ is true by Lemma D.3 and union bound. Now we assume the induction hypothesis holds for all levels $H, \ldots, h + 1$ and prove it holds for level $h$.

**Induction Hypothesis 1.** From Induction Hypothesis 2 for level $h+1$, it follows that value returned by Explore$(P(s,a))$ is in

$$\left[V^*(P(s,a)) - \frac{H-h}{H}\delta_r, V^*(P(s,a)) + \frac{H-h}{H}\delta_r\right]$$

for all $a \in \mathcal{A}$. Then, Induction Hypothesis 1 follows from Lemma D.3 and union bound.

**Induction Hypothesis 2.** It suffices to show that for any state $s \in \mathcal{S}_h$, when Line 8 in Explore$(s)$ is executed, then for all actions $a \in \mathcal{A}$

$$|f(s,a) - Q^*(s,a)| \le \frac{\rho}{2}. \tag{26}$$

Similar to proof of Theorem 5.1, we get

$$\sum_{(s_i,a_i,y_i)\in Y} |f(s_i,a_i) - y_i|^2 \le |Y|(\delta + \delta_r)^2. \tag{27}$$

We split the analysis in two cases:

1. we consider actions for which we execute Line 4 and

2. we consider rest of the actions.

**Case 1:** We now prove Equation (26) for all actions $a$ for which we execute Line 4. Similar to proof of Theorem 5.1, we get

$$|f(s,a) - Q^*(s,a)| \le \sqrt{18 \dim_E(\mathcal{F}, \rho/4)}(\delta + \delta_r) + \delta_r \le \frac{\rho}{2}. \tag{28}$$

**Case 2:** We now prove this for rest of the actions $a$. Similar to proof of Theorem 5.1, we get

$$\sum_{(s_i,a_i,y_i)\in Y} |f^*(s_i,a_i) - f(s_i,a_i)|^2 \le 4|Y|(\delta + \delta_r)^2. \tag{29}$$

Also, since we did not add this action to $Y$, by the definition of the oracle (Definition 5.1), we get

$$|Q^*(s,a) - f(s,a)| \le \frac{\rho}{2}, \tag{30}$$

which completes the proof. $\qquad\square$

**Lemma D.2.** *For any constant $c > 1$, if*

$$\rho \ge 4(\delta + \delta_r)\sqrt{\frac{c\dim_E(\mathcal{F}, \rho/4) - 1}{c - 1}} + 2\delta, \tag{31}$$

*then*

$$|Y| \le c\dim_E(\mathcal{F}, \rho/4). \tag{32}$$

*Proof.* Let $Y = \{(s_1, a_1, y_1), \ldots, (s_n, a_n, y_n)\}$. Similar to proof of Lemma 5.2, we can upper bound for any state-action pair $(s_j, a_j) \in \{(s_1, a_1), \ldots, (s_n, a_n)\}$, the number of disjoint subsequences $K$ in $\{(s_1, a_1), \ldots, (s_{j-1}, a_{j-1})\}$ that $(s_j, a_j)$ is $(\frac{\rho}{2} - \delta)$-dependent on, i.e.

$$K \le \frac{(j-1)(2(\delta + \delta_r))^2}{(\frac{\rho}{2} - \delta)^2}.$$

Also, for any sequence of state-action pairs say $\{(s_1, a_1), \ldots, (s_n, a_n)\}$, there exists a $(s_j, a_j)$ which is $(\frac{\rho}{2} - \delta)$-dependent on at least $\frac{n}{\dim_E(F, \frac{\rho}{2} - \delta)} - 1$ disjoint subsequences in $\{(s_1, a_1), \ldots, (s_{j-1}, a_{j-1})\}$. Therefore,

$$\frac{n}{\dim_E(F, \frac{\rho}{2} - \delta)} - 1 \le K \le \frac{(j-1)(2(\delta + \delta_r))^2}{(\frac{\rho}{2} - \delta)^2}. \tag{33}$$

That is, for any $\rho$ and $c > 1$ such that

$$\rho \geq 2 \left( 2(\delta + \delta_r) \sqrt{\frac{c \dim_E(\mathcal{F}, \rho/4) - 1}{c - 1}} + \delta \right), \tag{34}$$

we get

$$n \leq c \dim_E(\mathcal{F}, \rho/4). \tag{35}$$

$\square$

A simple concentration bound gives the following lemma:

**Lemma D.3.** *For any fixed state $s$ and action $a$, consider $n \geq \frac{H^2}{2\delta_r^2} \log \frac{1}{p}$ random independent samples $\{r_i(s, a)\}_{i=1}^n$ of random variable $R(s, a)$ with expectation $\bar{r}(s, a)$ and $r_i(s, a) \in [0, 1]$. Then,*

$$\left| \frac{1}{n} \sum_{i=1}^n r_i(s, a) - \bar{r}(s, a) \right| \leq \frac{\delta_r}{H}$$

*with probability at least $1 - p$.*