[Reviews · NeurIPS 2020]

Review 1

Summary and Contributions: This work describes a Q-learning with function approximation algorithm and provides an upper bound on the samples required to converge to the optimal policy in deterministic episodic MDPs. Specifically, assuming Q\* can be approximated up to some error \delta with approximation family F, the proposed algorithm converges to best policy linear in the eluder dimensionality of F O(dim_E) trajectories. The proof depends on an assumption of the size of the optimality gap \ro, which (informally) is the difference of the return between the optimal action and the second best. The upper bound holds for some error \delta, which is dependent on \ro. The proof complements a lower bound found in previous work, providing a tight bound.

Strengths: Despite significant advances in RL thanks to function approximation, little is known or proven about their theoretical properties. Better theoretical understanding and insights in guarantees are crucial to both further development and application of said RL. The authors provide an upper bound on the sample efficiency of arbitrary approximation functions, and hence should be important to the majority of the RL community. The paper is well written and provide both a specific example for linear function approximation and the more general case of arbitrary functions. Even a reader unfamiliar with the topic (e.g. myself) was able to follow most of the key arguments in the derivation and proof. Lastly, this work is closely related to the current literature which, to the best of my knowledge, are well documented and described.

Weaknesses: The proof, as described by the authors themselves, depend on the assumption on the gap optimality. The relationship between the approximation error and this optimality gap is crucial, a larger approximation error requires a larger gap to ensure the favorable properties. It is not entirely clear whether these bounds are meaningful in practice. Secondly, the algorithm for the general case requires an oracle to determine the most uncertain action given a state for the approximation family F. While it is argued that a similar oracle is used in previous work, it is not clear whether this is more realistic than previous work dismissed by the authors in related work ("Know-What-It-Knows" oracle in Li et al. 2011). The proof applies only to deterministic systems, restricting its application significantly. Note that similar proofs for more general stochastic MDPs seem out of reach for now and unrealistic at the moment.

Correctness: Unfortunately, I do not have the background to be qualified to guarantee the correctness. However, the proofs and argumentation looks rigid and thorough and I found no red flags.

Clarity: The paper is well written and, especially given the math-heavy content, is easy to follow.

Relation to Prior Work: This work is nicely put in the current literature, including insights in how it answers open questions posed in Wen and Van Roy 2013 and complements a proof by Du et al. 2002.

Reproducibility: Yes

Additional Feedback: * The range of the return, as described in the background section, is assumed to lie in [0,1]. This puts a strong bound on the values \ro can realistically take. Does this have any implication on the provided method? * The document goes into little detail into how to go about designing the required oracle described with equation (1). This oracle seems to be dependent on chosen F, is there a realistic scenario in which it is implementable? ---- Post response ---- Thank you for addressing my questions in the response, they clarified some of my misunderstandings.


Review 2

Summary and Contributions: This paper makes significant contributions to theoretical reinforcement learning by giving a nearly complete characterization of the setting where the environment is deterministic and the optimal $Q$-function can be approximated by a function in a function class with bounded Eluder dimension. The optimality is measured in terms of the approximation error and the sample complexity. This paper gives a tight phase transition on the approximation error: if the approximation error is smaller than a threshold depending on the optimality gap and the Eluder dimension, then the agent can learn the optimal policy efficiently; otherwise, the agent requires an exponential number of samples. The algorithm proposed in this paper also achieves a near-optimal sample complexity bound. These results resolve an open problem raised in [Wen and Van Roy, 2013].

Strengths: 1. Theoretical understanding of RL with function approximation is a very important topic. The setting studied in this paper is natural as many RL environments are deterministic, and bounded Eluder dimension is a standard assumption in the bandits/RL literature. 2. The theoretical results in this paper are technically strong. A complete theoretical characterization is rarely seen in the literature. This paper gives nearly complete characterizations in terms of the approximation error and the sample complexity. The results in this paper are conceptually interesting as there is a phase transition. 3. The recursion-based algorithm in this paper is intriguing and may inspire new algorithms.

Weaknesses: 1. There is a log factor gap in the bounds. 2. No experimental experiments are given in this paper, though I think it's fine for a theory paper.

Correctness: Yes.

Clarity: Yes.

Relation to Prior Work: Yes.

Reproducibility: Yes

Additional Feedback:


Review 3

Summary and Contributions: The paper studies reinforcement learning exploration with minimal assumptions on the Q* values, and misspecification of order the gap. It focuses on best policy identification as opposed to finding an approximate policy. It shows connection with the Eluder dimension. I find the paper interesting but there are several weaknesses that may make the result uninteresting.

Strengths: - connection with Eluder dimension - based on reasonable oracles (maximum uncertainty) - tight analysis for the setting they consider

Weaknesses: - Deterministic systems seem highly restrictive - The algorithm can do best policy identification only (it cannot find a near-optimal policy) - The model has to be very correct, i.e., misspecification smaller than roughly the gap - Cannot operate in the fairly common scenario where there exists more than one optimal policy.

Correctness: I believe the result is correct

Clarity: The paper is very well written indeed

Relation to Prior Work: The paper certainly cites most of the related literature. However, they should cite the weakness as well when comparing to this literature.

Reproducibility: Yes

Additional Feedback:

[Author Response · NeurIPS 2020]

We thank all the reviewers for their valuable feedback and appreciating our contributions. We first address some
common concerns.

**The proof applies only to deterministic systems / Deterministic systems seem highly restrictive.**  Despite deter-
ministic systems seem restrictive in theory, in practice, lots of RL problems are indeed deterministic. Moreover,
all known algorithms that work under the assumption that the optimal $Q$-function is linear require deterministic or
near-deterministic systems [Wen and Van Roy, 2013, Du et al., 2019].

**The proof depends on the assumption on the gap optimality / The model has to be very correct.**  In this paper,
we show that unless the gap $\rho = \Omega(\sqrt{\dim_E}\delta)$ where $\delta$ is the approximation error, any algorithm requires exponential
number of samples even just to find a near-optimal policy (see Proposition 1.2). Therefore, such an assumption is
necessary for any algorithm with polynomial sample complexity.

Please find our response to each individual reviewer below.

—— **To Reviewer #1** ——

**The algorithm for the general case requires an oracle.**  When the number of actions is finite (as in Atari games),
the agent can possibly enumerate all actions and find $f_1, f_2 \in \mathcal{F}$ separately for each action by running continuous
optimization algorithms that can handle constraints (e.g. projected gradient ascent). When the action space is continuous
(as in control tasks), the agent could directly optimize $a, f_1, f_2$ by running continuous optimization algorithms (as done
in practice). Moreover, we would like to note that our paper is concerned with the statistical efficiency, and the oracle
does not require any new sample (it solves an optimization problem based on existing samples).

Compared to the "Know-What-It-Knows" oracle, our uncertainty oracle just requires solving an optimization problem,
while it is even unclear whether the "Know-What-It-Knows" oracle can be implemented statistically efficiently for
general function classes. We will make the comparison clearer in the next version.

**The range of the return is assumed to lie in [0,1].**  This is a standard regularity assumption in RL theory, and is
required to make sure that the empirical mean of the reward values concentrates around their expectation by taking
enough samples. Such assumption is required for the algorithm in the supplementary material (Section D). In general,
if the summation of the reward values is in $[0, C]$, then the sample complexity of the algorithm in Section D will be
increased by a factor of $C^2$. Note that the required assumption that $\rho = \Omega(\sqrt{\dim_E}\delta)$ keeps unchanged even if one
changes the range of the reward values. E.g., if one scales all reward values by a factor of $C$, then the ratio between $\rho$
and $\delta$ remains unchanged.

—— **To Reviewer #2** ——

We would like to thank the reviewer for the positive feedbacks.

—— **To Reviewer #3** ——

**Cannot operate in the scenario where there exists more than one optimal policy.**  We disagree that our algorithm
does not work in the scenario where there exists more than one optimal policy. Consider the case that for some state $s$,
there are three actions $a_1$, $a_2$ and $a_3$. If $Q^*(s, a_1) = Q^*(s, a_2) = 1$ and $Q^*(s, a_3) = 0$, then by Definition 3.1, the gap
would be 1 and our algorithm still works. However in this case, it is clear that there could be more than one optimal
policy.

[Meta-Review · NeurIPS 2020]

This paper makes progress on our theoretical understanding of function approximation in RL, a crucial and tricky topic. The paper is technically strong and proposes a highly novel recursion-based algorithm that could open the door to future innovations.